# Limited impact of fingolimod treatment during the initial weeks of ART in SIV-infected rhesus macaques

Maria Pino[1], Amélie Pagliuzza[2], M. Betina Pampena [3], Claire Deleage [4], Elise G. Viox[1], Kevin Nguyen[1], Inbo Shim[1], Adam Zhang[1], Justin L. Harper [1], Sadia Samer[1], Colin T. King[1], Barbara Cervasi[5], Kiran P. Gill[5], Stephanie Ehnert[6], Sherrie M. Jean [6], Michael L. Freeman [7], Jeffrey D. Lifson[4], Deanna Kulpa [1,8], Michael R. Betts [3], Nicolas Chomont [2], Michael M. Lederman[7] & Mirko Paiardini [1,8] ✉

Antiretroviral therapy (ART) is not curative due to the persistence of a reservoir of HIV-infected cells, particularly in tissues such as lymph nodes, with the potential to cause viral rebound after treatment cessation. In this study, fingolimod (FTY720), a lysophospholipid sphingosine-1-phosphate receptor modulator is administered to SIV-infected rhesus macaques at initiation of ART to block the egress from lymphoid tissues of natural killer and T-cells, thereby promoting proximity between cytolytic cells and infected CD4+ T-cells. When compared with the ART-only controls, FTY720 treatment during the initial weeks of ART induces a profound lymphopenia and increases frequencies of CD8+ T-cells expressing perforin in lymph nodes, but not their killing capacity; FTY720 also increases frequencies of cytolytic NK cells in lymph nodes. This increase of cytolytic cells, however, does not limit measures of viral persistence during ART, including intact proviral genomes. After ART interruption, a subset of animals that initially receives FTY720 displays a modest delay in viral rebound, with reduced plasma viremia and frequencies of infected T follicular helper cells. Further research is needed to optimize the potential utility of FTY720 when coupled with strategies that boost the antiviral function of T-cells in lymphoid tissues.

Antiretroviral therapy (ART) successfully suppresses plasma viremia in the large majority of people living with HIV (PLWH). However, with very rare exceptions, ART withdrawal results in renewed viral replication and viral rebound; thus, lifelong antiviral therapy is needed.

To date, scalable strategies able to achieve drug-free immune control of HIV are not available. Recent efforts to achieve HIV eradication or a functional cure are being directed towards supplementing ART with immunotherapies targeting the rebound competent viral reservoir

[1]Division of Microbiology and Immunology, Emory National Primate Research Center, Emory University, Atlanta, GA, USA. [2]Centre de Recherche du CHUM and Department of Microbiology, Infectious Diseases and Immunology, Université de Montréal, Montreal, QC, Canada. [3]Department of Microbiology, Perelman School of Medicine, University of Pennsylvania, Philadelphia, PA, USA. [4]AIDS and Cancer Virus Program, Frederick National Laboratory for Cancer Research, Leidos Biomedical Research, Inc., Frederick, MD, USA. [5]Flow Cytometry Core, Emory Vaccine Center, Emory University, Atlanta, GA, USA. [6]Division of Animal Resources, Emory National Primate Research Center, Emory University, Atlanta, GA, USA. [7]Center for AIDS Research, Division of Infectious Diseases and HIV Medicine, Department of Medicine, Case Western Reserve University/University Hospitals Cleveland Medical Center, Cleveland, OH, USA. [8]Department of Pathology and Laboratory Medicine, Emory University School of Medicine, Atlanta, GA, USA. ✉e-mail: mirko.paiardini@emory.edu

(RCVR) that is established during acute HIV infection and maintained during ART. The RCVR consists of a pool of infected cells, including latently infected cells, containing replication competent proviruses that persist despite long term ART and can restart viral replication if ART is interrupted. Specifically, follicular helper T cells (Tfh), a subset of helper T cells that reside within the lymph nodes (LN) B cell follicles (BCF) and express high levels of CXCR5 and PD-1, have been described as a key cellular compartments for persisting HIV/Simian immunodeficiency virus (SIV) replication[1–5] that expand during chronic HIV/SIV infection[1,6]. A higher frequency of Tfh cells contains HIV/SIV-RNA than do extrafollicular CD4+ T cells in viremic PLWH and in SIV-infected rhesus macaques (RMs)[2,7,8], and are a major compartment of viral persistence in aviremic, ART-treated individuals[9,10]. Interestingly, Fukazawa et al. demonstrated that, distinguishable from progressor RMs, elite controller RMs with potent SIV-specific CD8+ T cell responses restrict SIV production to CD4+ Tfh cells within the BCF[3] suggesting that these infected cells are somehow protected from potent antiviral immune defenses. Multiple factors are thought to contribute to preferential HIV/SIV persistence in Tfh cells, including the presence of follicular dendritic cells (FDC) that can capture and efficiently transmit HIV to Tfh and the fact that most cytolytic CD8+ T cells, including HIV/SIV specific CD8+ T cells, do not generally traffick to the BCF in part based on lack of CXCR5 expression. While some studies have demonstrated that a subset of HIV/SIV-specific CD8+ T cells upregulate CXCR5 and are able to penetrate the BCF after vaccination or HIV/SIV infection[11–14], multiple groups have shown that CD8+ T cells largely fail to accumulate in BCF in both acute[15] and chronic SIV infection[8,13,16]. It is still debated whether the CXCR5+CD8+ T cells located within the BCF maintain killing capacity[17,18].

A key homeostatic mechanism for control of circulating lymphocyte levels involves regulation of the egress of T cells from LN into the circulation by a concentration gradient of sphingosine-1-phosphate (S1P)[19,20]. S1P is released by lymphatic endothelial cells and binds to S1P receptors (S1P1, S1P4, and S1P5) expressed on lymphocytes favoring their egress from lymphoid sites[21–23]. Interestingly, an impairment in T cell egress from LN through decreased S1P responsiveness has been observed in LN from untreated viremic PLWH that is restored with ART introduction[24]. Based on the above findings, developing therapeutic strategies aimed at improving the quantity and function of cytolytic cells in the LN is considered an approach of interest to limit the long-lived RCVR.

FTY720 (or fingolimod, analog of sphingosine) acts as a functional antagonist by binding to S1P1 and S1P3-5 receptors, inducing their internalization and proteasomal degradation, consequently inhibiting the binding to S1P and blocking the migration of lymphocytes to circulation[20]. FTY720 is clinically approved for the treatment of multiple sclerosis (Gilenya®, Novartis) where it is used to induce a profound reduction of circulating lymphocytes, attributed to the FTY720-induced retention of lymphocytes in LN[19,25,26]. Furthermore, we recently showed that treatment with FTY720 in aviremic, ART-suppressed SIV-infected RMs is safe, and effectively retained cytolytic immune cells in LN, and partially limited the frequency of Tfh harboring SIV during ART[27]. However, a significant fraction of Tfh were still infected and FTY720 did not affect the frequency of infection of other memory CD4+ T cell subsets. Based on our previous work, we hypothesized that activity of FTY720 to facilitate reduction of persistent virus might be limited by viral latency and the low number of SIV-specific T cells that can recognize and kill infected cells in animals that had been aviremic after several months of ART.

To follow up on these observations, we investigated the effect of FTY720 administered beginning at the time of ART initiation in SIV-infected RMs, a time when CD4+ T cells expressing viral antigens and HIV-specific cytolytic T cells are both found in larger numbers than seen in ART-suppressed animals. We hypothesized that such treatment might facilitate the interaction between cytolytic T cells and natural killer (NK) cells with infected CD4+ T cells in critical sites of viral replication and persistence such as the lymphoid tissues[28]. Thus, we determined whether FTY720 administered with early ART (i) is effective in trapping T cells and NK cells in lymphoid tissues and whether this results in; (ii) increased cytolytic clearance of SIV infected cells; and (iii) reduced number of infected cells that persists during ART.

## Results

### FTY720 treatment during the initial weeks of ART induces a profound peripheral lymphopenia in SIV-infected macaques

Twenty-two Indian RMs were infected intravenously (i.v.) with 300 TCID$_{50}$ SIVmac$_{239}$ and, at 6 weeks post-infection (wpi), started a triple combination of ART consisting of tenofovir disoproxil fumarate (TDF, 5.1 mg/kg per day), emtricitabine (FTC, 40 mg/kg per day) and dolutegravir (DTG, 2.5 mg/kg per day) combined in a daily subcutaneous injection. Fourteen RMs received only ART as control group (Gr. 1), while 8 RMs also received FTY720 at 500 µg/kg, orally, daily for 8 weeks (58 days), starting at ART initiation (Gr. 2; early FTY720 group). The dose of 500 µg/kg was chosen based on previous studies where FTY720 treatment in ART-suppressed RMs at 500 µg/kg daily for 4 weeks was shown to be safe[27]. Animals were assigned to experimental groups to balance characteristics including sex, weight, peak viremia and viral load setpoint just before initiation of ART. ART was continued for 51 weeks and was interrupted at 57 wpi. Six animals from the control group and the 8 FTY720 treated animals were followed for 4 months post analytic treatment interruption (ATI) to determine whether early FTY720 administration affected viral rebound and control. Blood and LN biopsies were collected longitudinally, and necropsies were performed at the end of the study (Fig. 1a).

To determine the effect of FTY720 on circulating lymphocyte numbers, we first determined by complete blood count and flow cytometry the absolute number and percentage frequency of circulating CD3+, CD4+, and CD8+ T cells before and after treatment. Consistent with its mechanism of action and our previous data in virologically suppressed animals[27], FTY720 significantly reduced the number of circulating T cells when compared with findings in RMs receiving only ART (Fig. 1b–d) both at 4 and 8 weeks of treatment, with the number of CD3+ T cells being significantly reduced from 831.2 ± 379.2 cells/µl (average ± SD) at baseline to 216.8 ± 228.8 cells/µl at week 8 in FTY720-treated animals, but increasing from 984.2 ± 516.1 cells/µl at baseline to 1718 ± 798.4 cells/µl at 8 weeks post ART initiation in controls (Fig. 1b). Both CD4+ and CD8+ T cell absolute numbers were significantly reduced in FTY720 treated animals at week 8 of treatment (to 8.5 ± 6.6 and 150.5 ± 122.6 cells/µl, respectively) as compared to ART-only controls at the same time point (928.9 ± 498.8 and 652.6 ± 262.9 cells/µl, respectively), with the number of CD4+ T cells being more profoundly affected than were CD8+ T cells (Fig. 1c, d). As such, while the frequency of blood CD3+ (of total lymphocytes) and of CD4+ T cells (of total CD3+ T cells) decreased during FTY720 treatment (Fig. 1e, f), the proportion of CD8+ T cells (of total CD3+ T cells) in this smaller total population increased (Fig. 1g). Importantly, FTY720 treatment during initiation of ART and for 8 weeks did not result in any recognized toxicities, as per veterinary observations and serum chemistry panels, and frequencies of T cells returned to normal levels at 1 month after FTY720 withdrawal (Fig. 1e–g), confirming that FTY720 effects were well-tolerated and temporary. The few CD4+ and CD8+ T cells present in blood during FTY720 treatment were enriched, as frequencies, for memory (CD95+) T cells with an activated (HLA-DR+) and cycling (Ki-67+) phenotype as compared with control RMs (Supplementary Fig. 1a–f; representative staining in Supplementary Fig. 1g) and, as expected, absolute counts of these cell populations were significantly reduced during FTY720 treatment when compared with control RMs (Fig. 1h–m). Consistent with their memory and activated status, the frequencies of circulating CD8+ T cells expressing the cytolytic molecules granzyme B and perforin were significantly

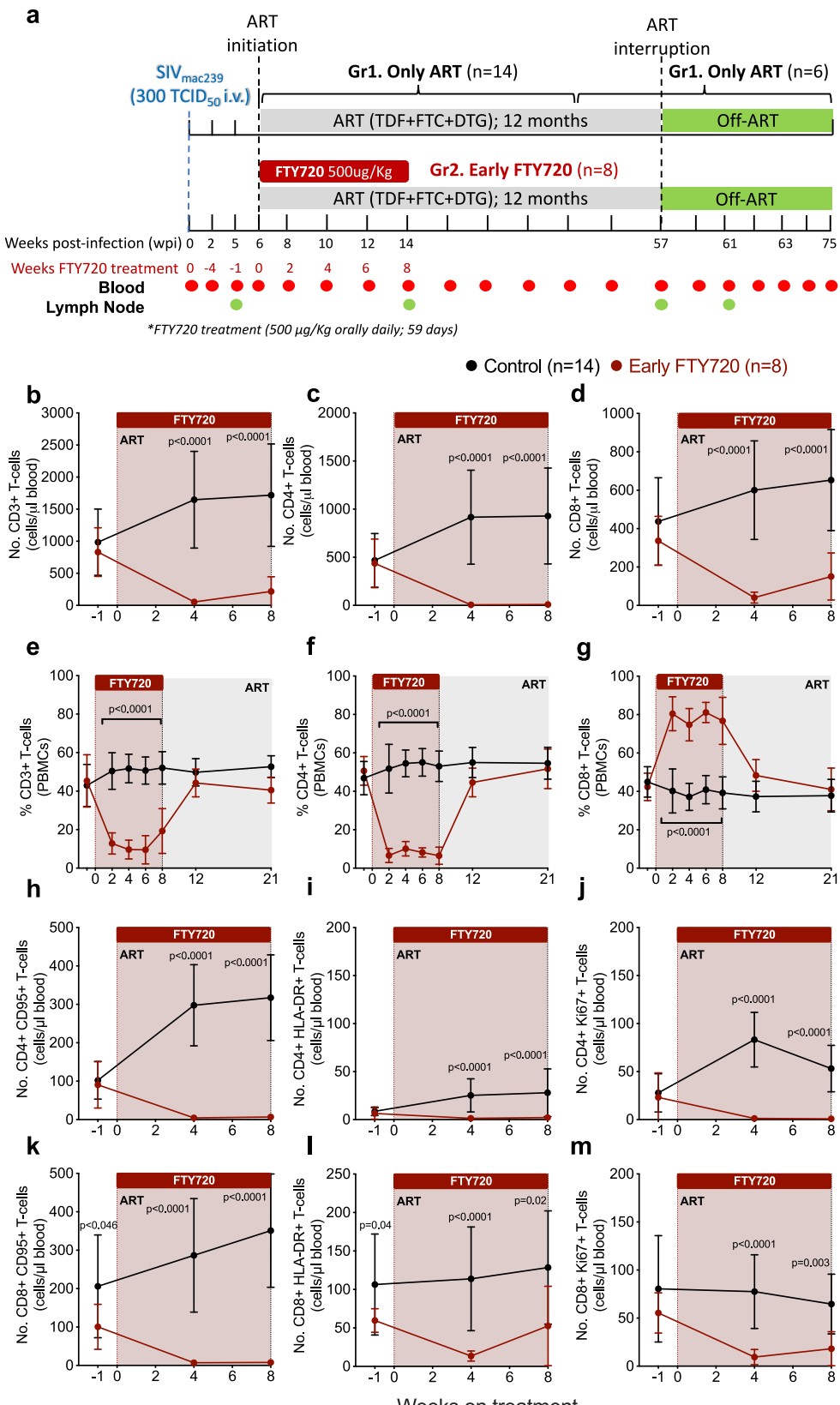

higher in FTY720 treated animals than among controls, although their absolute numbers remained low throughout the treatment (Supplementary Fig. 2a–d). We also observed a reduction in absolute counts of circulating B cells (CD3−HLA-DR+CD20+) and a slight reduction in NK cells (CD3−HLA-DR−CD20−NKG2A+CD8+), and monocytes (CD3−CD20−HLA-DR+CD14+) (Supplementary Fig. 2e–g).

To determine whether the changes induced by FTY720 treatment in the first 8 weeks of ART resulted in any persistent changes in cells of the immune system during long-term ART, we analyzed by flow cytometry the absolute counts and frequencies of T cell populations at the latest on-ART experimental points, up to 10 months after the last dose of FTY720. At those latest time points, absolute counts and

**Fig. 1 | FTY720 treatment in the first weeks of ART induces lymphopenia in SIV-infected RMs. a** Schematic of the study design. Twenty-two animals were infected with 300 TICD$_{50}$ SIVmac$_{239}$ and started antiretroviral treatment (ART) at week 6 post-infection (wpi). Animals were distributed into two groups: Gr1 received only ART (control group, $n = 14$ RMs), and Gr2 received 8 weeks of FTY720, starting at ART initiation (early FTY720, $n = 8$ RMs). Eight animals belonging to the control group were assigned to another study at ART interruption. **b–d** Absolute blood cell counts (cells/μl blood) and **e–g** frequencies of peripheral blood mononuclear cells (PBMCs) for **b**, **e** CD3+, **c**, **f** CD4+, and **d**, **g** CD8+ T cells from early FTY720 ($n = 8$ RMs) or control ($n = 14$ RMs) SIV-infected RMs. Absolute blood cell counts (cells/μl blood) for **(h–j)** CD4+ or **(k–m)** CD8+ T cells with a **(h, k)** memory (CD95+), **i, l** activated (HLA-DR+), and **(j, m)** cycling (Ki67+) phenotype in early FTY720 (red) or control (black) SIV-infected RMs. ART antiretroviral treatment. Data are presented as the mean ± SD. Statistical differences between FTY720 treated and control groups are indicated in asterisks and were assessed with a two-sided (95% CI) Mann–Whitney *U*-test.

frequencies of circulating CD3+, CD4+, and CD8+ T cells were not significantly different between untreated and FTY720 treated animals (Supplementary Fig. 3a–f), and no differences were observed between treatment groups when analyzing different T cell subsets (Supplementary Fig. 3g–l).

Thus, FTY720 administration in the first weeks of ART did not lead to significant adverse effects and induced a profound and transient reduction of circulating T cells, maintaining in blood only a small number of memory, activated, and cycling T cells. Changes in circulating cells were rapidly restored to levels similar to those in the control ART-treated group after FTY720 withdrawal.

### FTY720 treatment in the first weeks of ART increases the frequency of perforin-producing CD8+ T cells and the cytolytic potential of NK cells in LN

Next, we determined how FTY720 treatment at ART initiation affects the relative proportions of lymphocytes in LN. In the control group flow cytometry analysis showed that the frequency of CD4+ T cells was increased and that of CD8+ T cells was decreased 8 weeks after ART initiation as compared to pre-ART (week −1) (Fig. 2a, b). The proportional frequency of CD4+ and CD8+ T cells remained stable at the same time points in the FTY720 treated group, resulting in a significantly lower proportion of CD4+ T cells (Fig. 2a) and CD4+ to CD8+ T cell ratio (Fig. 2c) at week 8 on-ART as compared to findings in controls. These data suggest a preferential retention of CD8+ T cells in the LN of RMs treated with FTY720.

Next, we quantified by flow cytometry the fraction of LN CD8+ T cells expressing the cytolytic markers perforin and granzyme B at pre- (−1 week) and on-ART (8 weeks) time points (representative staining in Supplementary Fig. 1g). At the pre-ART time point, both study groups presented similar levels of CD8+ T cells expressing perforin or granzyme B (Fig. 2d, e). The frequency of CD8+perforin+ T cells increased during FTY720 treatment, resulting in significantly higher levels compared to controls at 8 weeks after ART initiation (Fig. 2d). Otherwise, the frequency of CD8+granzyme B+ T cells decreased in both treatment groups as expected due to a reduction in viral antigen, but to a slightly lower extent in the FTY720 treated RMs compared with control RMs (Fig. 2e). Similar results were observed when we analyzed levels of CD8+ T cells co-expressing granzyme B and perforin, with a statistically significant more pronounced reduction in the frequency of those cells in the control group (3-fold-reduction between weeks −1 and 8 on ART) as compared to the animals receiving FTY720 (1.5-fold-reduction between weeks −1 and 8 on ART and FTY720; Fig. 2f, g). Notably, the frequency of LN CD8+CXCR5+ T cells decreased during ART in controls but slightly increased with FTY720, thus resulting in significantly higher levels in the treated group as compared to controls at 8 weeks after ART initiation (Fig. 2h). Differently from total CD8+ T cells, FTY720 treatment did not impact on the frequency of granzyme B+perforin+ CD8+ T cells or NK cells expressing CXCR5.

To determine whether the forced retention in LN of CD8+ T cells induced by FTY720 treatment was associated with changes in their cytotoxic capacity, we performed a redirected killing assay[17] in purified CD8+ T cells. Briefly, P815 mastocytoma cells were coated with anti-CD3 monoclonal antibody, labeled with live/dead and cell tracer dye, and then co-cultured with different ratios of CD8+ T cells purified from

the LN of controls or treated RMs at 8 weeks of FTY720 treatment. CD8+ T cell cytolytic activity, measured by flow cytometry as the frequency of target cells expressing active caspase-3 (representative staining, Fig. 2i), was similar between control and FTY720-treated animals at different target to effector ratios (Fig. 2j).

To further analyze the effect of early FTY720 in accumulating cells in the LN, we then quantified by flow cytometry, the frequencies of NK cells (and their CD56 and CD16 subsets), B cells, and monocytes. We did not identify any difference in the frequency of B cells, monocytes or total NK cells (Supplementary Fig. 4). However, we observed a significant increased frequency of NK cells with a phenotype associated with cytolytic functions (CD16+CD56−; representative staining in Fig. 2k) in the LN of FTY720-treated RMs as compared to controls LN at 8 weeks of FTY720 treatment (mean ± SD: 22.3 ± 4.4 vs. 12.8 ± 5.2%; Fig. 2l), despite similar baseline levels (−1 week of FTY720 treatment: mean of 21.1 ± 5.5% in treated animals vs. 24.2 ± 8.4% in controls; Fig. 2l). Furthermore, and again despite similar levels at baseline (−1 week: mean fluorescence intensity of 1739 ± 570.6 in treated animals vs. 1792 ± 463.8 in controls; Fig. 2m), FTY720 treated animals presented significantly higher levels of CD16 in CD3−NKG2A+CD8+CD16+CD56− cells at 8 weeks of treatment, as assessed by mean fluorescent intensity (MFI; 3763 ± 895.7 in FTY720 treated animals vs 2177 ± 993.9 in untreated, Fig. 2m), indicating that FTY720 treatment not only resulted in a higher frequency of CD16+ NK cells, but also a higher expression of CD16 on a per cell basis, well-characterized features of cells with cytolytic activity[29].

Thus, FTY720 administration during the first 8 weeks of ART increases the frequency of total and perforin-producing CD8+ T cells and enhances the cytolytic phenotype of NK+ cells in the LN. Notably, the extent of those in vivo changes on CD8+ T cells were not sufficient to affect their ex-vivo killing activity on a per cell basis, as measured with a redirected killing assay.

### FTY720 administration in the first weeks of ART does not affect the size of the intact SIV reservoir

To determine if FTY720 administered during the first 8 weeks of ART affects the kinetics of viral decay, plasma viral loads (pVL) were measured longitudinally by qRT-PCR in the two groups of animals. Notably, pVL decay kinetics were nearly superimposable in the two groups (Fig. 3a), despite the very low blood CD4+ and CD8+ T cell counts in the FTY720-treated animals (Fig. 1b–d). This suggests that the initial decay of plasma viremia during ART is not primarily determined by the number of T cells circulating from lymphoid tissues to blood and that cell-free virus released from tissues, a process that is unaffected by FTY720, is likely the main contributor to pVL during ART. Both control and FTY720-treated animals successfully suppressed viremia below 60 copies/ml during ART and remained below this threshold for the duration of ART (Fig. 3a).

We previously showed that FTY720 treatment in SIV-infected RMs virally suppressed by ART reduced the cell-associated SIV-DNA content in the LN Tfh memory CD4+ T cell subset, but not in other CD4 T cell subsets[27]. To determine whether administration of FTY720 during the first 8 weeks of ART affected SIV-RNA and -DNA levels, we sorted by flow cytometry CD4+ memory (CD95+CD28+) T cells with a Tfh (PD-1+CD200hi) or non-Tfh (any other CD4+ memory T cell) phenotype from the LN of the animals in the study (sorting strategy showed in

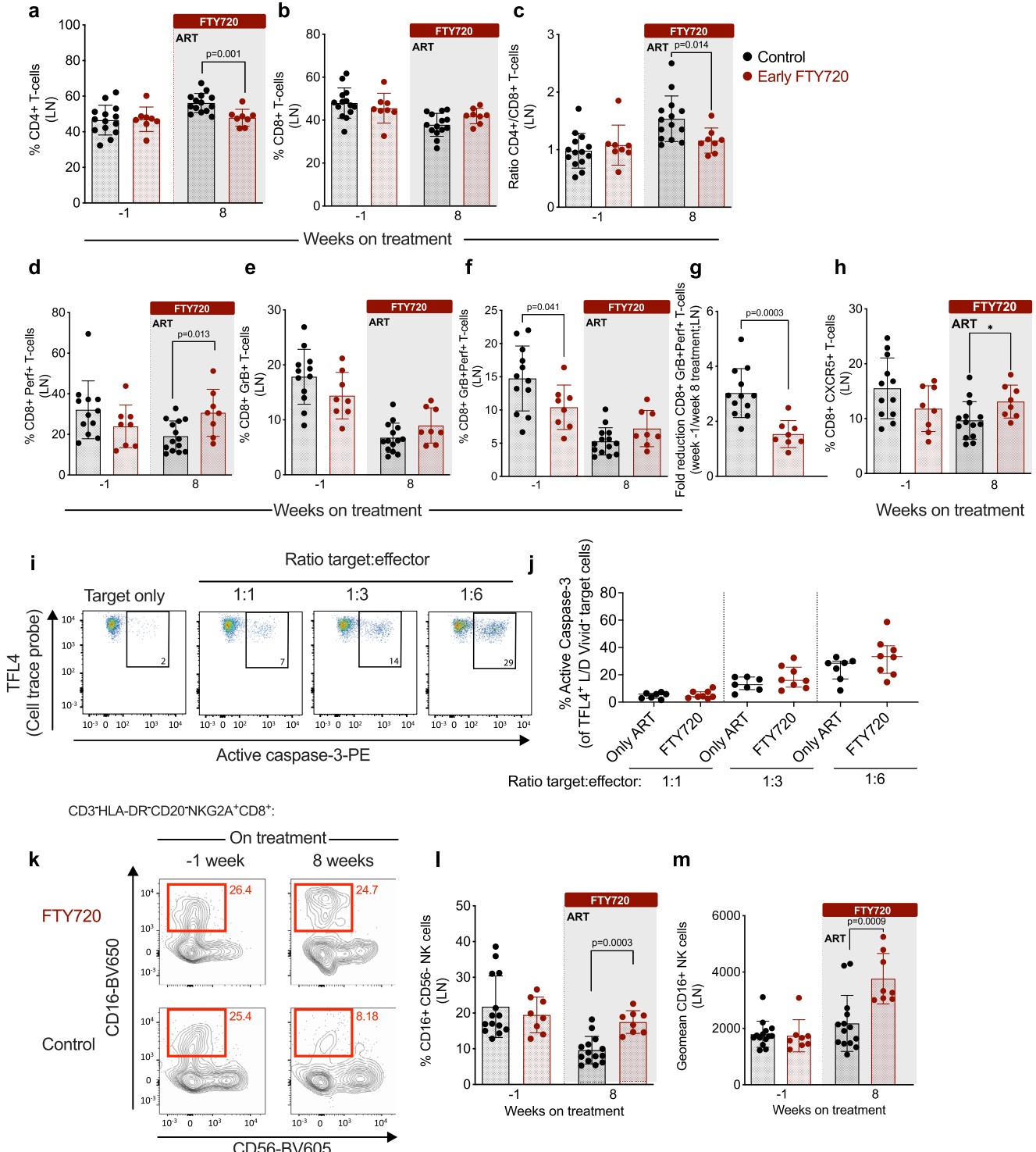

**Fig. 2 | FTY720 treatment in the first weeks of ART retains cytolytic cells but does not enhance their functionality in the LN. a** Frequency of CD4+ T cells, **b** CD8+ T cells, and (**c**) ratio of CD4+/CD8+ T cells in lymph node (LN) mononuclear cells at baseline (week −1) and at 8 weeks of treatment in early FTY720 (n = 8 RMs) or control (n = 14 RMs) SIV-infected RMs. **d** Frequency of LN perforin+ (Perf+), **e** granzyme B (GrB+), and (**f**) Perf+GrB+ CD8+ T cells at baseline (week −1) and at 8 weeks in early FTY720 (red) or control (black) SIV-infected RMs. **g** Fold reduction between week −1 and week 8 of Perf+GrB+ CD8+ T cells, and (**h**) frequency of LN CD8+ T cells expressing CXCR5 at baseline (week −1) and at 8 weeks in early FTY720 (n = 8 RMs) and control (n = 14 RMs) SIV-infected RMs. **i** Representative flow cytometry staining of P185 mastocytoma target cells (TFL4+ cells) expressing active

caspase-3 when cultured alone or co-cultured with different ratios of effector CD8+ T cells derived from LN. **j** Frequency of target cells expressing active caspase 3 following co-culture with CD8+ T cells derived from LN at week 8 from early FTY720 (n = 8 RMs) or control (n = 7 RMs) SIV-infected RMs. **k** Representative flow cytometry staining of CD56−CD16+NK+ cells in LN at week −1 and 8 from early FTY720 (top) or control SIV-infected RMs (bottom). **l** Frequency of CD56−CD16+ NK cells and (**m**) CD16 geometric mean fluorescence intensity (geomean) in NK cells derived from LN at week −1 and 8 from early FTY720 (n = 8 RMs) or control (n = 14 RMs) SIV-infected RMs. ART antiretroviral treatment. Data are presented as the mean ± SD. Statistical differences between FTY720 treated and control groups are indicated with *P* values and were assessed with a two-sided (95% CI) Mann–Whitney *U*-test.

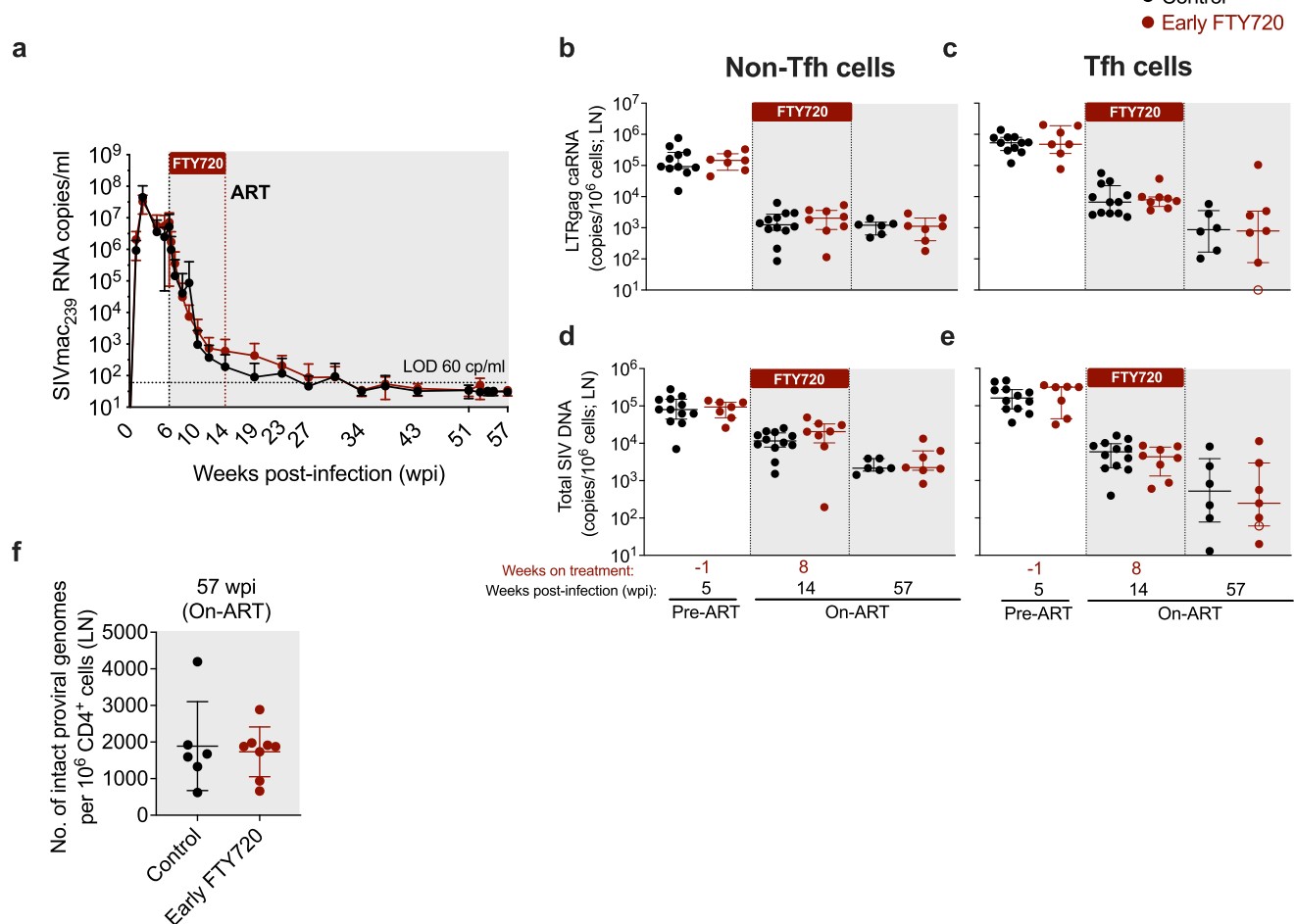

**Fig. 3 | FTY720 treatment in the first weeks of ART does not reduce plasma viremia or levels of cell-associated SIV-RNA or SIV-DNA in LN. a** Plasma viral loads (SIVmac$_{239}$ RNA copies/mL) measured by qRT-PCR longitudinally in early FTY720 or control SIV-infected RMs until 57 weeks post-infection (wpi). The dashed horizontal line represents the assay's limit of detection (LOD; 60 copies/mL) with undetectable events plotted as 30 copies/mL. **b, c** Cell-associated SIV-RNA (LTRgag caRNA; copies/10$^6$ cells) and (**d, e**) cell-associated SIV-DNA (copies/10$^6$ cells) in sorted (**b, d**) non-Tfh and (**c, e**) Tfh CD4+ memory T cells derived from LN of early FTY720 ($n = 8$ RMs) or control ($n = 14$ RMs) SIV-infected RMs at pre-ART (week −1 of treatment), on-ART (8 weeks of treatment), and at the last day on-ART (57 weeks post-infection, wpi). **f** Number of intact proviral genomes per 10$^6$ CD4+ T cells derived from LN of early FTY720 ($n = 8$ RMs) or control ($n = 6$ RMs) SIV-infected RMs at the last day on-ART (57 wpi). ART, antiretroviral treatment. Data are presented as the mean ± SD. Empty symbols in **c**, and **e** indicated undetectable values. Statistical differences between FTY720 treated and control groups were assessed with a two-sided (95% CI) Mann–Whitney U-test.

Supplementary Fig. 5). Levels of SIV-DNA and cell-associated RNA (LTR-gag transcripts) were determined by qPCR/qRT-PCR in these two subsets longitudinally, including at pre-ART/ pre-FTY720 treatment (5 wpi/−1 weeks of treatment); on-ART/8 weeks of FTY720 treatment (14 wpi); and last day on-ART (57 wpi). Cell-associated SIV-RNA and -DNA copies in sorted non-Tfh (Fig. 3b, d, respectively) and Tfh (Fig. 3c, e, respectively) CD4+ T cells were comparable between FTY720-treated animals and controls at all tested experimental points.

Finally, we quantified the levels of intact viral genomes in purified LN CD4+ T cells at the last day on-ART (57 wpi), using an intact proviral DNA assay (IPDA) specific for SIV[30–32]. No changes in the number of proviruses scored as intact by this assay were observed just prior to ART interruption between FTY720-treated animals and controls (Fig. 3f). Thus, FTY720 treatment in the first 8 weeks of ART limited neither the initial size nor the maintenance of persistent SIV during ART, including as measured by IPDA.

**A subset of FTY720 treated animals maintained lower plasma viremia and reduced SIV-DNA content in LN CD4+ Tfh cells early after ART interruption**

After 12 months of ART, and 10 months from the last dose of FTY720, ART was interrupted in 6 control animals and in the 8 FTY720

treated animals, with viral rebound followed longitudinally. The key immunologic and virologic parameters measured in the study were comparable between the 6 controls that underwent ATI and the other 8 animals in the control group (see "Methods: Study design," and Supplementary Fig. 6). Viral load rebounded to levels higher than 60 copies/mL in all (6 of 6) animals in the control group at 10 days off-ART, with an average plasma viremia of 2.8 ± 0.65 log$_{10}$ RNA copies/mL (Fig. 4a, b). In contrast, 4 out of 8 FTY720-treated animals maintained plasma viremia below 60 copies/mL at 10 days off-ART, resulting in a modest (4 days) but significant delay in viral rebound in treated RMs as compared to controls ($P = 0.048$; Fig. 4a). The 4 FTY720-treated RMs with undetectable viremia at day 10 rebounded with detectable levels at day 14 off-ART. Overall, post-rebound viral loads were similar when comparing the 6 controls vs. the 8 FTY720-treated RMs. However, the 4 FTY720-treated RMs with a delayed viral rebound had two-logs lower pVL than did the 6 controls at day 14 off-ART (2.45 ± 0.34 vs 4.43 ± 0.62 log$_{10}$ RNA copies/mL; Fig. 4b) and at least one-log lower pVL until day 43 off-ART (d20: 3.87 ± 0.32 vs 4.89 ± 0.6; d29: 3.3 ± 0.5 vs 4.3 ± 0.86; d37: 2.67 ± 0.97 vs 4 ± 0.96; d43: 3 ± 1.35 vs 4.2 ± 0.9 log$_{10}$ RNA copies/mL; 4 treated animals vs 6 controls). The improved control of viral rebound in a subset of FTY720 treated RMs was limited to the first 6 weeks post ATI, with

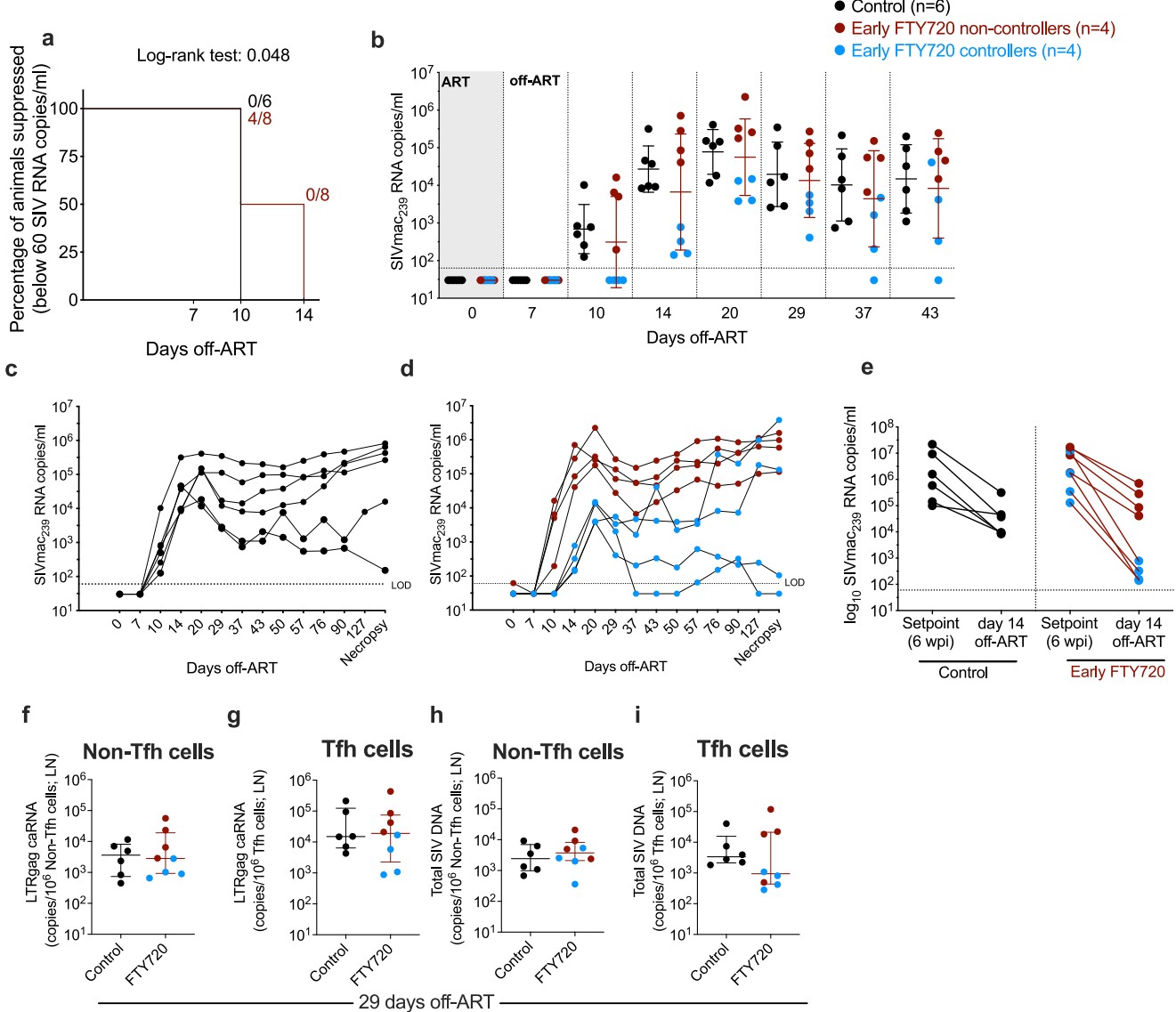

**Fig. 4 | A subset of FTY720 treated animals present reduced plasma viremia and Tfh infection in LN after ART interruption. a** Kaplan–Meier curve showing the percentage of animals maintaining plasma viral loads below 60 copies/ml in the first 2 weeks off-ART. **b** Plasma viral loads (SIVmac$_{239}$ RNA copies/mL) measured by qRT-PCR longitudinally after ART interruption in early FTY720 non-controllers ($n = 4$ RMs), early FTY720 controllers ($n = 4$ RMs), or control ($n = 6$ RMs) SIV-infected RMs. Plasma viral loads (SIVmac$_{239}$ RNA copies/mL) measured by qRT-PCR longitudinally in **c** control ($n = 6$ RMs), **d** early FTY720 non-controllers ($n = 4$ RMs), or early FTY720 controllers (n = 4 RMs) individual SIV-infected RMs after ART interruption. **e** Plasma viral loads (SIVmac$_{239}$ RNA copies/mL) of early FTY720 non-controllers ($n = 4$ RMs), early FTY720 controllers ($n = 4$ RMs), or control (n = 6 RMs) SIV-infected RMs at setpoint (pre-ART, 6 wpi) and at day 14 off-ART. **f, g** Cell-associated SIV-RNA (LTRgag caRNA; copies/$10^6$ cells) and (**h, i**) cell-associated SIV-DNA (copies/$10^6$ cells) in **f, h** non-Tfh and **g, i** Tfh CD4+ memory T cells from LN of early FTY720 non-controllers ($n = 4$ RMs), early FTY720 controllers ($n = 4$ RMs), or control ($n = 6$ RMs) SIV-infected RMs at 29 days off-ART. ART antiretroviral treatment. Data in **b** are presented as the mean ± SD. Data in **f–i** are presented as the mean ± interquartile range. Statistical differences in **a** between FTY720 treated and control groups were assessed with a log-rank test.

viral loads at later time points no longer significantly different from those of control RMs (Fig. 4c, d).

Parameters that have been associated with the timing and extent of SIV rebound, such as peak pVL during acute infection, pre-ART pVL (Fig. 4e), cell-associated SIV-RNA and SIV-DNA on-ART (shown in Fig. 3b–e) in the 4 FTY720-treated animals with better control of viral rebound were not significantly different when compared to those in untreated animals or the other FTY720-treated RMs.

Finally, we quantified levels of cell-associated SIV-DNA and RNA in LN-derived Tfh and non-Tfh CD4+ T cells at 29 days after ATI. Cell-associated SIV-RNA levels in non-Tfh (Fig. 4f) and Tfh (Fig. 4g) CD4+ T cell memory subsets did not differ between FTY720 treated and control animals. While SIV-DNA levels in non-Tfh cells were similar

between treatment groups (Fig. 4h), SIV-DNA in Tfh CD4+ T cells tended to be lower in the FTY720 treated group at 29 days off-ART, particularly for the subgroup of animals that had lower plasma viral levels throughout the off-ART follow-up (indicated in blue in Fig. 4i). Altogether, treatment with FTY720 for the first 8 weeks of ART, and terminated 10 months before ATI, was associated with a temporary reduced plasma viremia and frequency of SIV infected LN Tfh cells after ART interruption in a subset of FTY720-treated animals.

## Discussion

Cytolytic cells are normally relatively excluded from secondary lymphoid tissues[8,15,16,33], and do not recirculate due to the downregulation of lymphoid tissue homing receptors such as CD62L, CCR7, and CXCR5[34].

Here, we examined the potential utility of FTY720, a S1PR functional antagonist, for retaining infected CD4+ T cells and cytolytic CD8+ T cells in LN when administered for the first 8 weeks of ART in SIV-infected RMs as an approach to enhance immune surveillance and clearance of SIV infected cells from secondary lymphoid tissues. The rationale for treating at ART initiation is that both the number of SIV specific effector T cells and the number of CD4+ T cells expressing SIV is higher at this time than after months of ART, when both populations contract. We hypothesized that this strategy, by promoting close proximity of cytolytic CD8+ T and NK cells with CD4+ T cells, would have the potential to increase recognition and killing of HIV/SIV-infected cells and limit the establishment of the persistent RCVR. Furthermore, recent findings showed that the formation of the replication-competent HIV reservoir occurs around the time of therapy initiation[35]. As compared to ART-only controls, we found that 8 weeks of FTY720 treatment, started at ART initiation, was well tolerated, and did not result in any apparent adverse events. Compared to findings among ART only controls, FTY720+ART: (i) induced a profound lymphopenia while increasing the frequency of cytolytic CD8+ T cells and NK cells within the LN; (ii) did not increase pVL decay at ART initiation and did not reduce the establishment nor the persistence of the intact SIV reservoir during ART; and (iii) resulted in a modest delay of viral rebound (4 days) and a reduced SIV-DNA content in CD4+ Tfh cells in a subset of animals after ATI.

FTY720 administration at ART initiation induced a large and significant reduction in the numbers of blood T and B cells, with slight reductions in NK cells and monocytes. As previously reported in preclinical and clinical trials, this lymphopenia occurs largely consequently to lymphocyte sequestration in LN[36–39]. One of the main concerns when administering a drug that induces lymphopenia in the context of HIV infection is the potential risk of opportunistic infections or relapse of pre-existent latent infections; however, we did not observe evidence of any opportunistic infections among the FTY720 treated animals at the drug dose (500 μg/kg daily) given for 8 weeks. These results are aligned with our previous study where no toxicities were observed in ART-suppressed SIV-infected RMs that received the same dose of FTY720, but for a shorter period (4 weeks) and when the animals were already aviremic[27]. Furthermore, we showed that at one month after withdrawal of FTY720 treatment, circulating T cell frequencies and phenotype were reconstituted to levels similar to those of the ART-only control group, thus resulting in a limited duration of cytopenia that might place animals at risk. Another concern regarding FTY720 treatment is that, blocking lymphocyte egress from LN tissues could result in the accumulation of activated CD4+ T cells susceptible to HIV infection. This was not observed in Tfh or in non-Tfh CD4+ T cells when FTY720 was administered for the first 8 weeks of ART.

FTY720 increased the frequency of cells with cytolytic phenotype within the LN, including both CD8+ T cells producing perforin and CD16+ NK cells. We cannot discern if the failure of FTY720 treatment to limit the level of persistent virus measured by IPDA during ART is related to: an insufficient number of cytolytic cells; limited function of those cells; a limited number of target cells expressing viral antigens during ART; or a combination of those factors. In the RV254 cohort of individuals initiating ART in the earliest stages of acute HIV infection (Fiebig stages 1 and 2), HIV-specific CD8+ T cells differentiated into long-lived memory CD8+ T cells and had greater expansion and killing activity after peptide stimulation than those from individuals starting ART during chronic HIV infection. However, very early ART initiation leads to a lower number of HIV-specific CD8+ T cells than found among those who start ART during chronic HIV infection[40]. Thus, increasing the number of T cells present in tissue sites of viral replication and persistence of these cells in lymphoid tissues by FTY720 treatment could be particularly beneficial in the context of early ART when CD8+ T cell functions are preserved. Notably, in SIVmac$_{239M}$-infected RMs that initiated ART at peak plasma viremia (12 dpi) and were depleted of CD8+ T cells via anti-CD8β monoclonal antibody during ART

(10 weeks) and after ATI (6 weeks), CD8+ T cell depletion did not affect time to rebound or the number of rebounding SIVmac$_{239M}$ clonotypes after ATI, but resulted in a stable, approximately 2-log increase in off-ART set point plasma viremia relative to controls' levels[41]. This study suggests that while antiviral CD8+ T cell responses can develop and be maintained during ART, these responses are too low or too slow to limit the pace at which the SIV reservoir is reactivated and its initial spreading after ART cessation[41]. Interestingly, we also observed increased frequency of CD16+ NK cells within the LN upon FTY720 treatment. Previous studies in SIV-infected RMs reported that the levels of NKG2a/c$^{low}$CD16$^+$ terminally differentiated NK cells correlate with a reduction of replication competent SIV in PBMC during ART and with time to viral rebound following ATI[42,43].

Similarly, the increased frequency of T cells, including those expressing perforin, and NK cells within the LN at ART initiation among animals treated with FTY720 did not result in a faster decay of plasma viremia, suggesting that treatment did not increase the rate at which productively infected cells are eliminated in lymphoid tissue. These results indicate that other treatment interventions aimed at enhancing CD8+ T cell and NK cell functions, at redirecting them towards the BCF in the LN or at other sites such as the gut, or at increasing viral expression during ART, might increase effectiveness for reducing the SIV RCVR. Possible interventions of this nature might include combining FTY720 with IL-15 agonist, PD-1 blockade, or a potent latency reversing agent.

The finding that decay of plasma viremia during ART was unaffected by FTY720 despite profound reductions in blood lymphocyte counts indicates that plasma virus levels are not dependent on the number of blood CD4+ and CD8+ T cells, and suggests that the main source of plasma viremia is cell-free virus released from tissues. To the best of our knowledge, this is the first study assessing how interrupting T cell circulation from lymphoid tissues to blood at initiation of ART affects decay of plasma viremia and the size of the intact reservoir.

Following ART interruption, FTY720-treated animals had a modest delay in viral rebound compared with rebound in control animals. This delay was driven by a subset of FTY720-treated animals (4 out of 8 animals) that remained with undetectable plasma viremia at 10 days post ART while all control animals rebounded by day 10. This subset of animals had lower levels of plasma viremia during the first 6 weeks of ATI, and a reduced SIV-DNA content in CD4+ memory Tfh cells. These results are consistent with those of our previous study, where FTY720 treatment in ART-suppressed SIV-infected RMs resulted in lower SIV infection in Tfh CD4+ T cells, with the remaining CD4+ T cell subsets unaffected[27]. Since in animals in which plasma viral loads were already suppressed with ART FTY720 treatment limited SIV-DNA content exclusively in CD4+ Tfh cells, we sought to determine its efficacy when administered in the first weeks of ART in the presence of viral antigen, and by extension, more robust antiviral responses. When FTY720 was administered at ART introduction, we observed that the subset of animals with a delay in rebound of plasma viremia were those presenting lower SIV-DNA content in CD4+ Tfh at ATI.

This effect was selective as provirus levels measured by IPDA in total LN CD4+ T cells were comparable in treatment groups just before ART interruption, and even when the subset of treated animals that had lower plasma virus levels throughout the ATI were evaluated separately. We could not perform the IPDA on purified CD4+ Tfh as the number of available cryopreserved LN cells was insufficient for this analysis. Although Resop et al. recently reported that FTY720 treatment blocks cell-free and cell-to-cell transmission of HIV and reduces the establishment of latency in an in vitro primary cell model, no differences were observed in our in vivo study. One factor potentially accounting for these differences between the in vitro and in vivo models is the timing of FTY720 treatment. For example, in Resop et al. FTY720 was effective in reducing viral latency when added before HIV infection or before ART, but not if given following ART exposure.

Another difference is the presence of Vpx that can degrade the restriction factor SAM and HD domain-containing protein 1 (SAMHD1) in SIVmac239-infected RMs, which activation has been described in vitro as a mechanism of action for FTY720[44]. Thus, conceivably, FTY720 might be useful to limit seeding of the latent reservoir when started prior to ART initiation.

In conclusion, our study shows that effective retention of effector lymphocytes (CD8+ T cells and NK cells) within LN by FTY720 over the first 8 weeks of ART is not sufficient to affect SIV persistence in RMs during ART. However, it provides encouraging data on reducing plasma viremia and SIV infected Tfh cells in the absence of ART in a subset of animals. Future studies aiming at HIV remission could be directed towards treatments that synergize with the described FTY720 activity, including strategies that can increase numbers and function of cytolytic cells that can target HIV infected targets, while promoting latency reversal from a larger portion of the persistent virus pool.

## Methods

### Ethics statement
All animal experimentations were conducted following guidelines established by the Animal Welfare Act and by the NIH's Guide for the Care and Use of Laboratory Animals, Eighth edition. All procedures were performed in accordance with institutional regulations after review and approval by Emory University's Institutional Animal Care and Usage Committee (IACUC; Permit number YER2002876) at Emory National Primate Research Center (EPC). Animal care facilities are accredited by the U.S. Department of Agriculture (USDA) and the Association for Assessment and Accreditation of Laboratory Animal Care (AAALAC) International.

### Study design
Twenty-two specific pathogen-free (SPF) Indian RMs (*Macaca mulatta*) were housed at the Emory National Primate Research Center (EPC), Atlanta, GA, as recently described[27]. Animals were negative for Mamu-B*08, and Mamu-B*17, known protective alleles in the RM model of SIV infection[45,46]. All animals were infected intravenously with 300 TCID$_{50}$ of SIV$_{mac239}$ (provided by Koen Van Rompay, UC Davis) (Fig. 1a). Starting from week 6 post-infection (wpi) all animals were treated for approximately 12 months (up to 57 wpi) with a potent, combined ART that included tenofovir disoproxil fumarate (TDF; 5.1 mg/Kg per day), emtricitabine (FTC; 40 mg/Kg per day) and dolutegravir (DTG; 2.5 mg/Kg per day) formulated in a single subcutaneous daily injection (1 ml/Kg per day; s.c.)[47]. At ART introduction (6 wpi), the 22 animals were assigned to two treatment groups comparable in terms of age, weight, peak, and set-point viremia: untreated animals (control group, 14 RMs) and FTY720-treated animals (FTY720, 8 RMs). All animals in the cohort were female (Supplementary Table 1). FTY720 (acquired from Sigma-Aldrich) was administered orally once a day for the first 8 weeks of ART (from 6 to 14 wpi) at a dose of 500 μg/Kg per day, previously determined to be safe[27]. Animals continued on-ART for 12 months, and viral rebound was followed for additional 4 months after analytic treatment interruption (ATI). During this phase of the study, 8 animals from the untreated group were redirected towards a different study, for which they underwent additional immune interventions; thus, for the ATI phase, the study includes 6 control animals and 8 FTY720 treated animals. Of note, all the key immunologic and virologic parameters measured in the study were comparable between the 6 animals that underwent ATI and the other 8 controls that were redirected to a different study design (Supplementary Fig. 6). At the end of the ATI follow-up, all animals underwent necropsy.

### Sample processing
Collections of blood, and LN, were performed longitudinally during the entire study and at the necropsy. Blood samples were used for a complete blood count (CBC). Plasma was separated from EDTA-anticoagulated blood by centrifugation within 1 h of phlebotomy. From EDTA-anticoagulated blood, PBMCs were isolated using a Ficoll-Paque Premium density centrifugation (GE Healthcare), and washed with R10 media. R10 media was composed of RPMI 1640 (Corning) supplemented with 10% heat-inactivated fetal bovine serum (FBS, Corning), 100 IU/ml penicillin, 100 mg/mL streptomycin, and 200 mM L-glutamine (GeminiBio). For LN biopsies, the skin over the axillary or inguinal region was clipped and then surgically prepared. An incision was made in the skin over the LN, which was then exposed by blunt dissection and excised over clamps. LN biopsy was homogenized and passed through a 70-μm cell strainer to isolate lymphocytes and washed with R10 media. Mononuclear cells were counted for viability using a Countess II Automated Cell Counter (Thermo Fisher) with trypan blue stain. All samples were processed, stained, fixed (1% paraformaldehyde) and analyzed by flow cytometry within 24 hours of collection. Remaining mononuclear cells were cryopreserved and stored in liquid nitrogen for downstream assays.

### Determination of plasma viral load RNA
The number of SIVmac239 RNA copies per mL of plasma was quantified, as previously described, by quantitative reverse transcription polymerase chain reaction (RT-qPCR)[33] with a limit of detection of 15 copies/mL or, at few time points, by PCR[48] with a limit of detection of 60 copies/mL; therefore, the higher limit of detection was utilized for all data sets.

### Flow cytometric analysis
18-parameter flow cytometric analysis was performed on fresh PBMCs, and mononuclear cells (1 × 10$^6$ cells) derived from LN according to standard procedures using a panel of monoclonal antibodies (Abs) that we and others, have shown to be cross-reactive in RMs[49,50]. Abs were used as per manufacturer's recommendations: anti-CD95-PE-Cy5 (clone DX2, 10 μL, cat #559773), anti-Ki-67-Alexa Fluor 700 (clone B56, 5 μL, cat #561277), anti-CD56-BV605 (clone B159, 5 μL, cat. #740405), anti-CD16-BV650 (clone 3G8, 5 μL, cat. #563692), anti-CD3-BUV395 (clone SP34-2, 5 μL, cat #564117), anti-CD8-BUV496 (clone RPA-T8, 5 μL, cat #564804), and anti-CD14-BUV737 (clone M5E2, 5 μL, cat #564444), all from BD Biosciences; anti-CD4-APC-Cy7 (clone OKT4, 5 μL, cat #317418), anti-CD20-PerCP-Cy5.5 (clone 2H7, 5 μL, cat #302326), and anti-HLA-DR-BV711 (clone L243, 5 μL, cat #307644), all from Biolegend; anti-NKG2a-APC (clone Z199, 5 μL, cat #A60797) from Beckman Coulter; anti-GrB-PE-Texas Red (clone GB11, 5 μL, cat #GRB17) and Aqua Live/Dead Fixable Aqua from Invitrogen (AmCyan, 2 μL of 1:20 PBS dilution, cat. L34957); anti-Perforin-FITC (clone Pf-344, 5 μL cat #3465-7) from MABTECH. Flow cytometric acquisition was performed on at least 100,000 CD3+ T cells on a BD LSRII Flow Cytometer driven by BD FACSDiva software version 9.0. Analysis of the acquired data was performed by FlowJo software 10.8.1. (Tree Star Inc.).

### FACS cell sorting
Mononuclear cells isolated from LN were stained with anti-CD3-APC-Cy7 (clone SP34-2, 5 μL, cat #557757), anti-CD28-PE-CF594 (clone CD28.2, 5 μL, cat #562296), and anti-CD95-PECy5 (clone DX2, 10 μL, cat #559773) from BD Biosciences; anti-CD4-BV650 (clone OKT4, 2 μL, cat #317436), anti-PD-1-BV421 (clone EH12.2H7, 5 μL, cat # 3299220), and anti-CD200-PE (clone OX-104, 5 μL, cat #329206) from Biolegend; anti-CD8-FITC (clone 3B5, 5 μL, cat #MHCD0801-4) from Thermo Fisher Scientific; and Aqua Live/Dead Fixable Aqua from Invitrogen (AmCyan, 2 μL of 1:20 PBS dilution, cat. L34957). Sorting of CD4+ Tfh (PD1+CD200hi) was performed using a FACS AriaII (BD Biosciences) in samples collected before, during FTY720 treatment, at ART interruption, and 29 days after ART interruption. Post-sorting FACS analysis determined that sorted CD4+ T cell subsets were on average >96% pure.

## Quantifications of SIV-DNA and cell-associated SIV-RNA

DNA and RNA were simultaneously extracted from PBMCs or LNMCs using the Allprep universal kit (Qiagen) according to the manufacturer's instructions. Total SIV DNA (LTR-gag) and cell-associated RNA (LTR-gag) were quantified in all samples by nested quantitative PCRs as described previously[49], with minor modifications. Total SIV DNA was amplified in a first round of PCR with 2 primers that anneal within a conserved region of the long terminal repeat (LTR) 5′ end (SIVLLTRfwd: ATG CCA CGT AAG CGA AAC TGG CAG ATT GAG CCC TGG GAG) and at the junction with the Gag gene (SIVgagrev: TGC TGC CCA TAC TAC ATG CTT C). The forward primer SIVLLTRfwd was extended with a lambda phage−specific heel sequence at the 5′ end of the oligonucleotide. Primers targeting the CD3 gene (HCD3OUT-5′: ACT GAC ATG GAA CAG GGG AAG and HCD3OUT-3′: CCA GCT CTG AAG TAG GGA ACA TAT) were also added to quantify the exact number of cells in the initial samples. Gag-LTR sequences were amplified from 15 µL DNA in a 50-µL reaction mixture composed of 1× Taq Buffer, 3 mM MgCl2, 300 µM dNTP, 300 nM SIVLLTRfwd, 300 nM SIVgagrev and 2.5 U Taq polymerase (Invitrogen). The first-round PCR cycle conditions were as follows: a denaturation step of 8 m at 95 °C and then 12 cycles of amplification (95 °C for 1 m, 55 °C for 40 s, 72 °C for 1 minute), followed by an elongation step at 72 °C for 15 minutes. In a second round of PCR, the lambda T−specific primer (Lambda T: ATG CCA CGT AAG CGA AAC T) and the LTR primer (SIVLTRrev2: CTT TAA GCA AGC AAG CGT GGA G) were used to amplify SIV sequences obtained from the first amplification. Primers targeting CD3 were also used in another second-round PCR. Nested PCR was performed on one-tenth of the first- round PCR product in a mixture consisting of 1× Quantinova Probe PCR mix (Qiagen), 1250 nM Lambda T primer, 1250 nM SIVLTRrev2 primers, and 200 nM SIVLTRZen probe (GCA GGT AGA GCC TGG GTG TTC CCT GC). For CD3 amplification, nested PCR was performed in a mixture composed of 1X QuantiNova Probe PCR mix (Qiagen), 1250 nM HCD3IN 5′ (GGC TAT CAT TCT TCT TCA AGG T), 1250 nM MamuCD3IN 3′ (TAA GAT GGC GGT AAC AGG GT) and 200 nM MamuCD3ZEN probe (AGC AGA GAA CAG TTA AGA GGC TCC AT). Cycling was performed on the Rotor-Gene (Qiagen) with a denaturation step (95 °C for 4 min), followed by 40 cycles of amplification (95 °C for 3 s, 60 °C for 10 s). The total SIV DNA copy number was calculated using a standard curve as a reference. This standard curve consisted of serial dilution of the 3D8 cell lysates (carrying 1 integrated copy of SIV genome per cell). Cell-associated RNA were quantified using the same protocol with the following modifications: RT and pre-amplification were performed using the Superscript III RT/Platinum Taq Mix (Invitrogen) and CD3 primers were omitted. RT-PCR cycle conditions were as follows: A reverse transcription step of 30 min at 50 °C, followed by a denaturation step of 2 min at 94 °C and then 16 cycles of amplification (94 °C for 15 s, 55 °C for 30 s, 68 °C for 1 min), followed by an elongation step at 68 °C for 5 min. The second amplification was performed as described above for SIV DNA. To accurately measure the absolute number of copies of cell-associated SIV RNA, serial dilutions of in vitro transcribed RNA generated from plasmids using T7 polymerase were used. SIV RNA copies were normalized per million cells using the CD3 quantification measured in the DNA assay.

## Redirected killing assay

P815 mastocytoma target cells (ATCC TIB-64™) were labeled with LIVE/DEAD Fixable Violet (Thermo Fisher Scientific) and TFL4 (OncoImmun), washed twice in PBS, and incubated for 30 min at room temperature with α-CD3 (1 mg/mL; clone SP34-2; BD Biosciences). CD8+ T cells were negatively selected from blood and frozen mononuclear cells derived from LN using a NHP CD8+ T Cell Enrichment Kit (StemCell Technologies). Isolated CD8+ T cells were rested in complete medium for at least 45 min at 37 °C and then incubated with α-CD3-coated P815 cells at different effector to target ratios in a 96-well V-bottom plate for 4 hours at 37 °C. Cells were then stained with α-active caspase-3−PE (clone C92-605; BD Biosciences) and α-CD8−BV785 (clone RPA-T8; Biolegend) and acquired using a Symphony (BD Biosciences). Killing activity was calculated by subtracting the frequency of active caspase-3+TFL4+LIVE/DEAD− P815 cells in target-only wells from the frequency of active caspase3+TFL4+LIVE/DEAD− P815 cells in wells containing CD8+ T cells.

## Macaque CD4+ T lymphocytes enrichment and DNA extraction

LN biopsy-derived, cryopreserved mononuclear cells were removed from liquid nitrogen storage, and thawed in 37°C water bath. Next, cells were transferred to pre-warmed sterile RPMI media as soon as the ice core was dislodged. After counting and spinning down the cells, the cell pellets were resuspended in media contained DNase. Following positive selection of CD4+ T cells using the CD4+ T Cell Isolation Kit, non-human primate, Miltenyi Biotec (cat#130-0910102), larger-molecular-size (100-200 kb) DNA was isolated with the Gentra Puregene Cell kit (Qiagen) to minimize shearing of proviral DNA. Finally, DNA concentration and purity were identified by NanoDrop.

## ddPCR assay for intact SIV genomes

Intact Proviral SIV DNA assay was performed as previously described[31]. Briefly, 6 replicates of 500 ng DNA were added to the master mix containing 2× ddPCR Supermix for Probes (no dUTP, Bio-rad), 600 nM of each primer and 200 nM of each probe targeted regions of *pol* and *env*. Primer and probe sequences are listed in Supplementary Table 2. Droplets were generated by using the Droplet Generator QX200™ (Bio-rad). The droplets were typically left at 4 °C overnight or for at least 4 h to cool off before reading by QX200 Droplet Reader. Input cell numbers were quantitated by rhesus macaque gene (*RPP30*). The two regions of *RPP30* were designed ~11 kbp apart. Previous publications also compared the ratio of RPP30 single vs double positive droplets to estimate the DNA shearing index (DSI), as size of an integrated intact SIV proviral genome is approximately same length spanned by the two targeted regions of *RPP30*. Four replicates of 6 ng DNA were added to the master mix containing 2x ddPCR Supermix for Probes (no dUTP) (Bio-rad), 500 nM of each primer, and 250 nM of each probe. Details of primer and probe sequences and PCR conditions are included in Supplementary Table 3. The frequency of 2-LTR circles were measured by adapting the primers and probes from Policicchio's group[51]. Four replicates of 500 ng DNA were added to the master mix containing 2× ddPCR Supermix for Probes (no dUTP, Bio-rad), 600 nM of each primer and 200 nM of each probe targeted regions of *2-LTR* and *env*. The frequency *of 2-LTR⁺env⁺* circles was subtracted from the frequency of intact (*pol⁺env⁺*) proviruses.

## Statistical analysis

Data analyses were performed using GraphPad Prism version 9.4.1. (GraphPad Software, Inc., La Jolla, CA). The results are expressed as the mean ± SD. Statistical significance of immunophenotyping and viral data between time points and study groups were performed using a paired or Mann−Whitney unpaired $U$-test when appropriate. A $P$ value < 0.05 was considered statistically significant.

## Reporting summary

Further information on research design is available in the Nature Research Reporting Summary linked to this article.

# Data availability

The raw data for all graphs generated in this study are provided in the Supplementary Information/Source Data file. Source data are provided with this paper.

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

## Acknowledgements

We gratefully acknowledge Gilead Sciences (Romas Geleziunas), and ViiV Healthcare (Chris Parry and Katie Kitrinos) for supplying the antiretroviral drugs. The authors also thank all the animal care and veterinary staff at the Emory National Primate Research Center (EPC) for providing animal and veterinary care. The SIVmac$_{239}$ strain used to infect the RMs was kindly provided by Koen Van Rompay (UC Davis). We thank Thomas Vanderford and Shelly Wang at the Emory Center for AIDS Research (CFAR) Virology and Molecular Biomarkers Core for determining plasma viral loads at selected time points. This research is supported by NIH R33AI116171 (to M.M.L. and M.P.); NIH P01AI131338 (to M.R.B.); ERASE HIV UM1AI164562 (to M.P.), co-funded by National Heart, Lung and Blood Institute, National Institute of Diabetes and Digestive and Kidney Diseases, National Institute of Neurological Disorders and Stroke, National Institute on Drug Abuse and the National Institute of Allergy and Infectious Diseases; NIH Office of the Director, Office of Research Infrastructure Programs, P51OD011132 and U42OD011023 to EPC, and Center for AIDS Research at Emory University P30AI050409 and in part with federal funds from the National Cancer Institute, National Institutes of Health, under Contract Nos. 75N91019D00024 and HHSN261201500003I. The content of this publication does not necessarily reflect the views or policies of the Department of Health and Human Services, nor does mention of trade names, commercial products, or organizations imply endorsement by the U.S. Government. The funders had no role in study design, data collection and analysis, decision to publish, or preparation of the manuscript.

## Author contributions

M.P., J.L.H., M.L.F., M.M.L., and M.P. contributed to conceptualization. J.D.L. contributed to plasma viral quantification. M.P., A.P., C.D., I.S., M.B.P., A.Z., K.N., S.S., J.L.H., C.T.K., B.C., and K.P.G. contributed to data curation. M.P., A.P. C.D., M.B.P., and A.Z. contributed to formal analysis. M.M.L., and M. Paiardini contributed to funding acquisition. M.P., M.B.P., S.E., S.M.J., J.L.H., N.C., D.K., M.B., M.M.L., and M.Pa. contributed to investigation. M.P., A.P., C.D., E.G.V., I.S., M.B.P., A.Z., K.N., S.S., J.L.H., C.T.K., S.E., and S.M.J. contributed to methodology. J.L.H. contributed to project administration. D.K., M.B., N.C., M.M.L., and M. Paiardini contributed to supervision. M.P., A.P., C.D., M.B.P, A.Z., I.S., B.C., and K.P.G. contributed to validation. M. Pino and M.B.P. contributed to visualization. M.P., M.L., M.P. contributed to writing original draft. All authors contributed to manuscript development and have critically reviewed and approved the final version.

## Competing interests

The authors declare no competing interests.
