## [Peer review file · Nature Communications]

REVIEWER COMMENTS

Reviewer #1 (Remarks to the Author):

In this manuscript by Pino et al, the authors described the cellular and virologic effects of early treatment of SIV-infected rhesus macaques with Fingolimod or FTY720. This study complements a recent study from the same group characterizing the effects of FTY720 in ART-suppressed SIV-infected macaques. For this particular study, macaques were treated daily with FTY720 at the same time as ART initiation for 8 weeks. Characterization of cellular subsets in blood is done prior to ART initiation, while FTY720 treatment and 13 weeks after stopping FTY720. Similar studies are done in lymph nodes prior to initiation of treatment and at week 8 of FTY720 treatment. Finally, the authors evaluated levels of viremia during treatment and after ART interruption. The major finding of this study is a slight delayed of viral rebound in some of the animals treated with FTY720. In general, this study highlights the potential of using FTY720 during early treatment to modulate the latent reservoir and demonstrate the safety of using this immunomodulator in the context of SIV infection. Albeit the study found limited benefit of FTY720 while administered at the same time of ART, it highlights the potential of using early interventions to modulate the latent reservoir. There are some concerns with the manuscript.

Major concerns

- In the evaluation of cell subsets in blood upon FTY720 treatment, the authors found a decrease in total numbers of CD4 and CD8T cells but an increase in certain subsets. In particular, they observed an increased in memory, activated and cycling T-cells as % cells in PBMCs. Absolute numbers instead of % as in Fig 1B may be more informative to evaluate whether these cell subsets are insensitive or less sensitive to FTY720. Do the authors have any explanation why these cell subsets are not affected by FTY720?
- In Figure 2, the authors evaluate changes in the % of CD4 and CD8 in LN upon FTY720. These changes seem to be minimal. Are there changes in the number of cells? That should be more informative to address that the decrease in cell number observed in blood is associated with an increase in cell number in LN and not any other off-target effects of FTY720.
- The authors discuss the effects observed by Resop et al of FTY720 inhibiting HIV viral infection and the establishment of latency in vitro. This results are not confirmed in this SIV in vivo study. One of the effects of FTY720 in blocking HIV infection is associated with the activation of the restriction factor SAMHD1. This may not be recapitulated with SIV due to the presence of Vpx, which degrades SAMHD1. The authors should expand their discussion to explain the differences in the studies and why FTY720 or associated pathways may be still useful to limit the seeding of the latent reservoir in HIV infection.

Minor

- The authors indicate that the experimental groups were balance based on sex. However, there is no information provided in the number of male/female macaques used in the study and whether sex could have any effect in the biological observations.

- Line 80. Extra comma

- Line 265. Add space in "toART"

- Line 307. Add space in "viruswhen"

Reviewer #2 (Remarks to the Author):

The authors in this manuscript are reporting on the ability of Fingolimod treatment to reduce or delay virus rebound in rhesus macaques following ART interruption. The design of the study is very well thought off. The experiments and the controls were very clearly presented. This is a very well written paper. The authors discussed all the caveats and explained their results objectively. It was disappointing that the effect of Fingolimod treatment was marginal and ineffective in limiting viral rebound or delays given their previous publication (ref 27). Even though Fingolimod was ineffective in vivo, I think the paper provided novel findings that worth reporting.

Major concerns

1: I am not sure how to interpret the data from ref 27 (Pino et al) and the current data. In the paper by Pino et al there was a strong decrease in integrated virus so why this did not translate to reduction and delay of virus rebound following cessation of ART. I think the authors need to discuss this further and compare these 2 studies.

2: The title does not reflect the data and is very optimistic. I would not say limit (very strong word) since there was only a delay of 4 days.

3: The NK data looks very promising and more discussion on the relevance of NK cells in killing virally infected cells.

4: Do the authors know whether at any point during Fingolimod treatment the retention of NK and T cells were in marginal zones. Do they express high levels of CXCR5 especially the CD8+ CTL.

Reviewer #3 (Remarks to the Author):

In a previously published paper, the authors demonstrate that the lysophospholipidsphingosine-1 phosphatereceptormodulator - FTY720 (fingoli-mod) effectively sequestered circulating CD4+ and CD8+ T cells in lymphoid tissues in SIV infected macaques treated with antiretroviral drugs. This treatment increased the number of CD3+ T cells in lymph nodes and reduced the circulated reservoir of SIV infected cells in circulating T cells as well as SIV infected T cells in lymphoid tissues.

In the manuscript under review the authors found that in SIV infected, ART treated macaques, FTY720 treatment macaques had severe loss of lymphocytes and an increased number of CD8+ t cells expressing perforin in lymph nodes. However, there was no change in the latent reservoir of SIV infected lymphocytes measured by the intact SIV genomes assay. In addition, when antiretroviral therapy was withdrawn from SIV infected controls and FTY720 treated macaques, rebound of SIV in blood occurred in 10 days in all of the untreated macaques and in 4/8 of the FTY720 treated macaques. SIV rebound occurred at 14 days post ART cessation in the remaining four FTY720 treated macaques.

This is a well-done study, however, the results clearly demonstrate that treatment with FTY720 has no significant impact on either the size of the SIV latent reservoir or on rebound of virus when ART is stopped. The lack of efficacy of FTY720 treatment greatly decreases the impact of this study. This reviewer does not support the publication of this manuscript.

We thank the Editor and Reviewers for their critical review of the manuscript and thoughtful comments. We believe that the revised manuscript, in which we have addressed all the main concerns, is substantially improved and suitable for publication in *Nature Communications*.

Please note that in the manuscript file we have highlighted in yellow the revised/new parts of the manuscript.

Reviewer #1

Major concerns

1. In the evaluation of cell subsets in blood upon FTY720 treatment, the authors found a decrease in total numbers of CD4 and CD8 T cells but an increase in certain subsets. In particular, they observed an increase in memory, activated and cycling T-cells as % cells in PBMCs. Absolute numbers instead of % as in Fig 1B may be more informative to evaluate whether these cell subsets are insensitive or less sensitive to FTY720. Do the authors have any explanation why these cell subsets are not affected by FTY720?

Answer: As per reviewer request, we have included in Figure 1d and 1e absolute count numbers of memory (CD95+), activated (HLA-DR+) and cycling (Ki67+) T-cells, and moved the corresponding frequency analysis to Supplementary Figure 1a, and 1b. Although frequencies of memory, activated and cycling T-cells were increased, absolute counts presented a significant reduction after introduction of FTY720 treatment. Thus, those cells are still sensitive to FTY720 treatment, although relatively less as compared to other cell subsets, likely due to their activated/effector phenotype, considering that effector cells downregulate CCR7 and CD62L, thus have a lower potential to home back to lymphoid tissues.

2. In Figure 2, the authors evaluate changes in the % of CD4 and CD8 in LN upon FTY720. These changes seem to be minimal. Are there changes in the number of cells? That should be more informative to address that the decrease in cell number observed in blood is associated with an increase in cell number in LN and not any other off-target effects of FTY720.

Answer: We concur with the reviewer that changes in the % of CD4 and CD8 T-cells in lymph node (LN) upon FTY720 treatment are limited. Notably, since FTY720 affects several cell populations, we do not expect large changes in CD4 and CD8 T-cells in terms of frequency. We also agree that an increase in the number of cells in LN would be more informative to address the decrease in cell number observed in blood. Importantly, in our previous study we showed by complex imaging analyses that most of ART-suppressed SIV-infected rhesus macaques treated with FTY720 presented a higher number of T-cells in LN that were collected during necropsy at the end of the treatment (Pino et al. 2019). However, obtaining enough LN tissue to reliably perform these type of analyses (in addition to the immunological and SIV reservoir quantification analyses) without sacrificing the animals during the treatment, as for the current study in which animals have been followed for months after FTY720 was terminated, is very challenging and was not possible in the current study. Finally, there are several important limitations when performing quantification of absolute cell numbers in LN, even when enough tissue is available. For example, it is plausible that an effective blockade of T cell egress will increase the size of the LN more than the cell density; thus, it may be inaccurate to quantify absolute cell counts in term of cell density without knowing the size of the entire LNs before and after FTY720 treatment.

3. The authors discuss the effects observed by Resop et al of FTY720 inhibiting HIV viral infection and the establishment of latency in vitro. These results are not confirmed in this SIV in vivo study. One of the

effects of FTY720 in blocking HIV infection is associated with the activation of the restriction factor SAMHD1. This may not be recapitulated with SIV due to the presence of Vpx, which degrades SAMHD1. The authors should expand their discussion to explain the differences in the studies and why FTY720 or associated pathways may be still useful to limit the seeding of the latent reservoir in HIV infection.

Answer: The two systems are very different, thus making challenging a direct comparison. Resop et al. showed that FTY720 treatment prior to HIV infection inhibited cell-free HIV infection, and FTY720 treatment after HIV infection and before ART introduction inhibited the establishment of latent reservoir; here, we performed the FTY720 treatment in animals that were already infected and with de novo infection inhibited by ART; our study was not designed to determine a direct role of FTY720 in inhibiting viral infection but in promoting elimination of infected cells. However, Resop et al. also performed an in vitro experiment where FTY720 and ART were introduced together after the HIV reservoir was established. Interestingly, in this experimental setting more similar to our in vivo study, they did not observed differences in reactivated HIV reservoir concluding that, in their model, latent infection is established prior to ART and following ART exposure the latent reservoir is not impacted by FTY720. As pointed by the reviewer, another important difference is the presence of Vpx that degrades SAMHD1 in SIV_{mac239}-infected rhesus macaque, which activation has been described as an important mechanism of action for FTY720 in Resop et al. As requested, we have expanded the discussion to compare our data with those by Resop et al. and we have discussed the potential role of Vpx and SAMHD1.

Minor

- The authors indicate that the experimental groups were balance based on sex. However, there is no information provided in the number of male/female macaques used in the study and whether sex could have any effect in the biological observations.

Characteristics of the animals (sex, weight, age, etc) has been added in Material and Methods, and new Supplementary Table 1. All animals in this cohort were female, so we could not assess if there were different study outcomes due to sex differences.

- Line 80. Extra comma;
- Line 265. Add space in “toART”;
- Line 307. Add space in “viruswhen”;

All fixed in the new revised manuscript.

Reviewer #2 (Remarks to the Author):

The authors in this manuscript are reporting on the ability of Fingolimod treatment to reduce or delay virus rebound in rhesus macaques following ART interruption. The design of the study is very well thought off. The experiments and the controls were very clearly presented. This is a very well written paper. The authors discussed all the caveats and explained their results objectively. It was disappointing that the effect of Fingolimod treatment was marginal and ineffective in limiting viral rebound or delays given their previous publication (ref 27). Even though Fingolimod was ineffective in vivo, I think the paper provided novel findings that worth reporting.

Answer: We thank the reviewer for the very positive comments on the quality of our study and manuscript, and to highlight the importance of reporting the message that Fingolimod, as monotherapy, is not sufficient to impact the size of the reservoir or the kinetics of viral rebound.

Major concerns

1: I am not sure how to interpret the data from ref 27 (Pino et al) and the current data. In the paper by Pino et al there was a strong decrease in integrated virus so why this did not translate to reduction and delay of virus rebound following cessation of ART. I think the authors need to discuss this further and compare these 2 studies.

Answer: As requested, we further expanded the comparison among the two studies in the discussion of the revised manuscript. Briefly, in the previous paper, where FTY720 was administered to ART-suppressed rhesus macaques, we did find a reduction in the SIV-DNA levels only on Tfh cells, with the large majority of the remaining reservoirs – including CM, TM, and EM CD4 T-cells– having levels of DNA comparable between FTY720 treated and control animals. Thus, it is not surprising that a difference limited to Tfh cells was not sufficient to significantly impact the time or the extend of viral rebound after ATI. Indeed, that was the rationale for changing the design and to use FTY720 at ART initiation in the current study, with the hope to have a bigger impact in reducing the reservoir that persist during ART.

2: The title does not reflect the data and is very optimistic. I would not say limit (very strong word) since there was only a delay of 4 days.

Answer: As suggested, we have changed the title to: Limited impact of fingolimod treatment during the initial weeks of ART on plasma viremia and Tfh cell infection in SIV-infected rhesus macaques.

3: The NK data looks very promising and more discussion on the relevance of NK cells in killing virally infected cells.

Answer: We thank the reviewer for this important point; we expanded on the role of NK cells in the discussion of the revised manuscript.

4: Do the authors know whether at any point during Fingolimod treatment the retention of NK and T cells were in marginal zones. Do they express high levels of CXCR5 especially the CD8+ CTL.

Answer: Similarly to what is described for the expression of perforin in CD8+ T-cells, when analyzing CXCR5 expression in CD8+ T-cells within the lymph node (LN) via flow cytometry, we observed that the frequency of LN CD8+ CXCR5+ T-cells decreased during ART in controls but slightly increased during FTY720 treatment, resulting in significantly higher levels compared to controls at 8 weeks after ART initiation. This new set of data has been included in the revised Fig. 2 and showed below. We also specified (data not shown) that FTY720 treatment did not impact on the frequency of CD8+Perf+GrB+ or NK cells expressing CXCR5 (reported below).

Reviewer #3 (Remarks to the Author):

In a previously published paper, the authors demonstrate that the lysophospholipid sphingosine-1 phosphate receptor modulator - FTY720 (fingolimod) effectively sequestered circulating CD4+ and CD8+ T cells in lymphoid tissues in SIV infected macaques treated with antiretroviral drugs. This treatment increased the number of CD3+ T cells in lymph nodes and reduced the circled reservoir of SIV infected cells in circulating T cells as well as SIV infected T cells in lymphoid tissues. In the manuscript under review the authors found that in SIV infected, ART treated macaques, FTY720 treatment macaques had severe loss of lymphocytes and an increased number of CD8+ t cells expressing perforin in lymph nodes. However, there was no change in the latent reservoir of SIV infected lymphocytes measured by the intact SIV genomes assay. In addition, when antiretroviral therapy was withdrawn from SIV infected controls and FTY720 treated macaques, rebound of SIV in blood occurred in 10 days in all of the untreated macaques and in 4/8 of the FTY720 treated macaques. SIV rebound occurred at 14 days post ART cessation in the remaining four FTY720 treated macaques.

This is a well-done study, however, the results clearly demonstrate that treatment with FTY720 has no significant impact on either the size of the SIV latent reservoir or on rebound of virus when ART is stopped. The lack of efficacy of FTY720 treatment greatly decreases the impact of this study. This reviewer does not support the publication of this manuscript.

Answer: We thank the reviewer for acknowledging this is a “well-done study”. We also agree that our study demonstrates that treatment with FTY720, by itself, does not impact on the size of the latent SIV reservoir and is not sufficient to control viral rebound when ART is interrupted. This was discussed in the original version of the manuscript, including proposing additional interventions that, based on our data, have the potential to synergize with FTY720. This has been further acknowledged in the revised manuscript, including revising the title. We respectfully disagree with the reviewer that the lack of efficacy of the treatment greatly decrease the impact of the study. There is a huge interest and agenda in the HIV cure field in implementing interventions that will improve the number and functions of cytolytic T cells in tissues and/or favor a close interaction between infected cells and cytolytic cells in tissues. By showing that this is not sufficient to impact the size of the reservoir and the kinetics of viral rebound is, in our opinion, a very important contribution. Furthermore, as indicated in the manuscript, this is the first study showing that the decay of plasma viremia during ART is completely independent by the number of circulating CD4 T cells.

REVIEWERS' COMMENTS

Reviewer #1 (Remarks to the Author):

The authors have addressed my previous concerns

Reviewer #2 (Remarks to the Author):

The authors have successfully responded to my critiques